# BlockLLM: Memory-Efficient Adaptation of LLMs by Selecting and Optimizing the Right Coordinate Blocks

## Abstract

Training large language models (LLMs) for pretraining or adapting to new tasks and domains has become increasingly critical as their applications expand. However, as model and data sizes grow, the training process presents significant memory challenges, often requiring a prohibitive amount of GPU memory that may not be readily available. Existing methods such as low-rank adaptation (LoRA) add trainable low-rank matrix factorizations, altering the training dynamics and limiting the model's parameter search to a low-rank subspace. GaLore, a more recent method, employs Gradient Low-Rank Projection to reduce the memory footprint, in the full parameter training setting. However GaLore can only be applied to a subset of the LLM layers that satisfy the "reversibility" property, thus limiting their applicability. In response to these challenges, we introduce BlockLLM, an approach inspired by block coordinate descent. Our method carefully selects and updates a very small subset of the trainable parameters without altering any part of its architecture and training procedure. BlockLLM achieves state-of-the-art performance in both finetuning and pretraining tasks, while reducing the memory footprint of the underlying optimization process. Our experiments demonstrate that BlockLLM achieves superior performance on finetuning both large and small models. On pretraining a Llama model on C4 dataset, BlockLLM is able to train with significantly less memory than the state-of-the-art, while still maintaining competitive performance.

## 1 Introduction

Recent advancements in natural language processing (NLP) have been propelled by the development of large language models (LLMs) Le Scao et al. (2023); Touvron et al. (2023); OpenAI (2023); Almazrouei et al. (2023). These models have set new benchmarks for a variety of NLP tasks, including language translation Takase & Kiyono (2021), text summarization Kedia et al. (2021), and sentiment analysis (Brown et al., 2020). The core strength of LLMs lies in their scale. Empirical evidence suggests that increases in model size not only enhance performance across standard benchmarks but also unlock new capabilities that are absent in smaller models (Kaplan et al., 2020; Zhao et al., 2023). Pretraining and finetuning LLMs on domain-specific application data have enhanced their applicability immensely.

However, pretraining and finetuning LLMs are resource-intensive processes and require substantial memory and computational power. For example, a $7B$ parameter Llama model demands approximately 14GB of memory Zhao et al. (2024), assuming each parameter is a 16-bit float occupying 2 bytes. The memory required for storing gradients during backpropagation is similarly substantial, adding another 14 GB. Additionally, LLMs are often trained using the Adam optimizer (Kingma & Ba, 2014) and its variants, which maintain first and second moment estimates for each parameter. This effectively doubles this memory requirement, resulting in an additional 28 GB of VRAM memory. Consequently, the total memory required for the weights, gradients, and optimizer states amounts to a substantial 56 GB. The effect of LLMs training's high memory requirement is far reaching, in that it comes down to the question of who can train these large models. With the memory calculations above, a 7 billion parameter model can only be trained on A100 GPUs or above. As models grow larger, this memory burden will continue to escalate, restricting large-scale LLM

training to only researchers and organizations with the most advanced GPUs. This is a significant barrier to entry for practitioners who do not have access to such high-end hardware.

**Existing strategies for memory-efficient training.**
To address these challenges, multiple strategies are being explored to reduce the number of parameters, gradients, and the corresponding optimizer state size. One popular strategy is the application of *pruning methods*, where a large set of parameters or entire layers are removed from the model architecture Wang et al. (2019); Ma et al. (2023); Sun et al. (2023). However, pruning approaches often require extensive retraining to recover lost accuracy Fan et al. (2019). Furthermore, identifying which parameters are crucial before training is challenging Michel et al. (2019); Sajjad et al. (2023). This challenge complicates implementation and can lead to generalization issues, particularly on diverse or unseen data Ma et al. (2023).

PEFT (*Parameter-Efficient-Fine-Tuning*) methods Hu et al. (2021); Lialin et al. (2023); Hu et al. (2023) achieve memory efficiency by introducing low-rank matrices to the transformer architecture. This significantly reduces the number of trainable parameters needed during fine-tuning. Although integrating these low-rank matrices alleviates the extensive retraining demanded by pruning techniques, they can alter the training dynamics. This could potentially lead to quality issues during the merging phase He

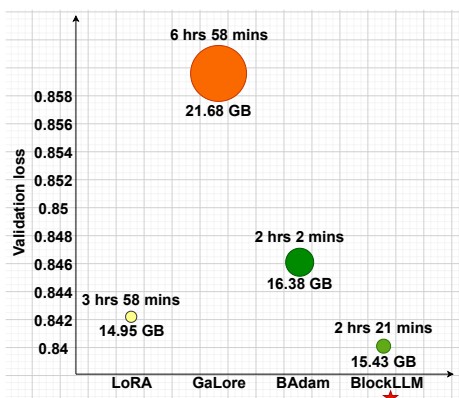

Figure 1: Illustration of validation loss, memory usage, and training time across various training methods for fine-tuning LLaMA-2 on the Alpaca dataset. BlockLLM demonstrates superior performance, achieving lower memory consumption and reduced training time.

et al. (2021). The low-rank assumption may also constrain the model's expressiveness, limiting its ability to fully capture complex patterns in the data. Furthermore, the additional parameters introduced by PEFT methods can increase the model's parameter size, countering efforts to reduce overall model size.

A recent work, *GaLore* Zhao et al. (2024) focuses on full parameter training and achieves memory efficiency by performing low-rank factorization of the gradients in specific layers. However, GaLore does not achieve high memory efficiency across all model types, as its gradient factorization method can only be applied to layers that satisfy the reversibility property. This limitation restricts its applicability and efficiency in models where not all layers exhibit this property.

Techniques such as gradient and activation checkpointing Chen et al. (2016), quantization Han et al. (2015), and parameter offloading Rhu et al. (2016) are also commonly used to achieve memory savings. However, these methods often come with trade-offs such as increased computational overhead or compromised performance. For instance, checkpointing Chen et al. (2016) reduces memory usage but requires re-computation, quantization lowers precision and can affect accuracy Han et al. (2015) and parameter offloading increases data transfer latency Rhu et al. (2016). While these methods have their own limitations, many of these techniques are complementary to the approach presented in this work and provides additional opportunities for memory reduction when used in combination.

**Block coordinate descent (BCD).** BCD is popular algorithm in the large-scale optimization literature. At any training iteration $t$, instead of updating all the parameters $W$, BCD updates only a block of parameters $b_t$ by setting $W_{t+1}^{b_t} = W_t^{b_t} + d_t$, where $d_t$ is the update for that block. Importantly, $b_t$ does not stay fixed across iterations. As a result, BCD does not constrain model performance, which is often the case with low-rank approximation methods. Moreover, BCD doesn't alter the model architecture in any way and preserves its structure throughout the training process. As it can be seen, this formulation directly falls in the reduced parameter training regime. This insight forms the cornerstone of our approach and hence the name BlockLLM.

A related study by (Belilovsky et al., 2019) demonstrated that sequentially solving one-hidden-layer problems could match the performance of large model training, inspiring us to tackle parameter

and memory efficiency by training large models in parts. While their approach loses the full model context by focusing on smaller sub-problems, it inspired us to investigate how maintaining the full model context during training could potentially yield even better results. Finally, the convergence of BCD has been theoretically proven on various problem architectures in the optimization literature (Ramesh et al., 2023; Zeng et al., 2019; Nutini et al., 2022; Nesterov, 2012; Richtárik & Takáč, 2014; Shalev-Shwartz & Zhang, 2013), which inspired us to adapt it to large-scale LLM training.

Some previous works have explored training various neural network models using BCD (Zeng et al., 2019; Lau et al., 2018; Massart & Abrol, 2022). A recent parallel study (Luo et al., 2024) extends BCD to LLMs, focusing on memory efficient training. These methods update one block of parameters per iteration, chosen either randomly or cyclically. This approach is often inefficient for large-scale models because, in most cases only a small subset of parameters requires to be updated during finetuning (see our analysis 2 for details). If these critical parameters are not updated frequently enough, training becomes prolonged, impacting overall efficiency. In this work, we address this issue by updating parameters that are important for training more frequently. Our experimental results (see Figure 1) demonstrate that this strategy leads to both better performance and more efficient training.

**Our Contributions.** The key contributions of this work are as follows:
1. We propose a novel parameter and memory-efficient algorithm, **BlockLLM** where we dynamically select and train a block of parameters. This approach minimizes memory consumption by maintaining gradient and optimizer states only for the selected parameters.
2. We introduce a novel block selection criterion tailored for LLM training where impactful parameters are updated more frequently. This leads to faster training, as important parameters are updated earlier in the process.
3. We demonstrate that BlockLLM achieves state-of-the-art training and generalization performance in both fine-tuning and pretraining tasks, while also enabling faster training and reduced memory usage.
4. We provide extensive ablation studies showing that BlockLLM is robust across various hyperparameter settings, including sparsity and patience.

## 2 METHODOLOGY

They key idea behind BlockLLM is to select and update only a subset of parameters during training, enabling us to achieve significant memory savings. An illustration of the method is given in Figure 2. However, the criteria for selecting the "right" subset of parameters is not clear. In this section, we look at magnitude pruning as a tool to identify parameter importance in the finetuning setting.

**Magnitude Pruning.** Magnitude pruning is a widely recognized technique for reducing the parameter count in neural network models Gupta et al. (2022). In this analysis, we use the weight magnitude of a parameter as a measure of parameter importance and study the impact of training on the selected parameters. First, we trained DistilBERT (Sanh et al., 2019) on the IMDb dataset (Maas et al., 2011) for sequence classification, achieving an accuracy of 92.02%. We then conducted inference on the GLUE-CoLA dataset (Wang et al., 2018) without fine-tuning, resulting in a significant drop in accuracy to 47.74%. This drop in performance, possibly due to domain shift, encouraged us to use this setup for our analysis.

Next, we performed magnitude pruning on the IMDb pretrained model at various sparsity levels. We then finetuned these pruned models on the GLUE-CoLA dataset. Let $s$ denote the sparsity level, $W^t$ represent the model parameters at iteration $t$, and $n$ be the total number of parameters. For each parameter $w_i$ where $i = 1, \ldots, n$, we compute $|w_i|$. During training, we update only $S = \text{Top}_k |W^0|$, where $k = n \times (1 - s)$. The results of these experiments, detailing the relationship between sparsity and accuracy, are summarized in Table 2.

Interesting results emerged from this analysis: at $0.5$ sparsity, the model retains a high accuracy of $78.52\%$, suggesting that up to $50\%$ of parameters can be pruned with minimal performance loss. However, accuracy drops significantly at $0.7$ sparsity to $67.68\%$. *This suggests that there is some inductive bias that the model can leverage when finetuning on a dataset with significant domain*

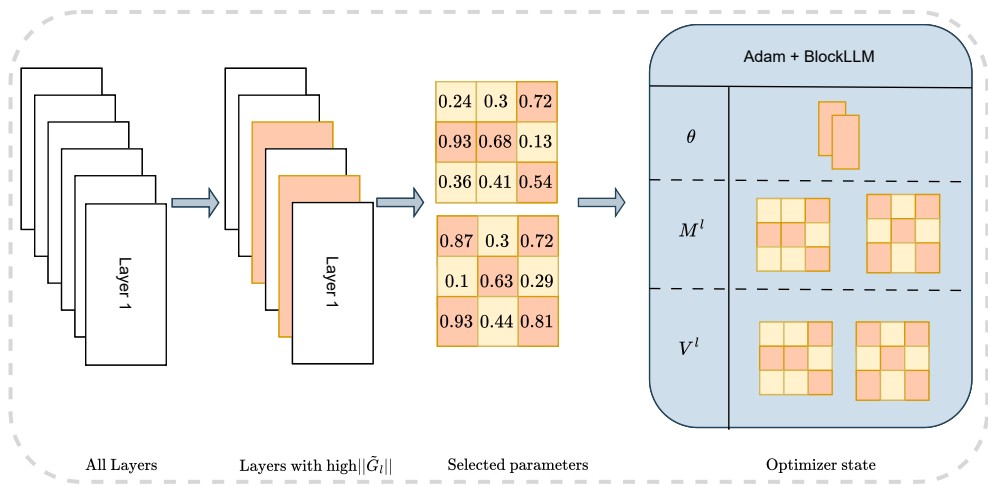

Figure 2: Given a large language model consisting of many layers, BlockLLM first finds the layers with the largest gradient $||\tilde{G}_l||$ (highlighted in orange) and selects a subset of the parameters. During optimization, only the selected layers will be updated ($\theta$) and the optimizer will keep track of its optimizer states only for those selected parameters (shown in orange).

*shift, under a reduced parameter setting.* However, estimating the sparsity $s$ apriori is hard, and its not clear what factors influence this.

**Analysis of Weight Magnitudes.** To further understand the effects of this parameter selection strategy, we analyzed $|W|$ before and after training. This helped understand which weights are updated more frequently and their impact on model performance. Specifically, we compared the initial weights $|W^0|$ and final weights $|W^t|$ of the model after fine-tuning on the CoLA dataset (Wang et al., 2018). Our findings are presented in Figure 3. The histogram on the left of Figure 3 shows $|w_i^t|$ for all $i$ where $\delta_i > \eta$. Here, $\delta_i = |w_i^0 - w_i^t|$ and $\eta$ is the threshold. The histogram on the right of Figure 3 plots the frequency of $\delta$ in the updates, revealing that a large percentage of the updates were minor.

Several additional insights and questions emerge from these observations. The histogram of $\delta$ indicates that most weight changes are minor, suggesting that significant updates are concentrated in a small portion of the weights. These experiments raise some pertinent questions:

1. Is it reasonable to judge the impact of a parameter during training based on its initial weight magnitude?
2. How should the appropriate sparsity level $s$ be determined before commencing training?
3. Although we observed that only a few parameters undergo significant changes during training, which specific parameters are updated during various phases of the training process?

### 2.1 ANALYSIS OF REDUCED PARAMETER TRAINING

To address the above questions, we further investigated the pruning process. First, we updated the chosen parameter set $S$ every $m$ iterations based on the weight magnitudes $|W^t|$, at current iteration $t$. This means that after every $m$, the parameter selection criteria is revisited to obtain a new set of parameters to update. The objective is to understand if $S$ changes significantly over time and if adaptively selecting $S$ enhances the training performance. In this framework, we continue to update only the top $k = n \times (1 - s)$ parameters in each iteration. However, the percentage of unique parameters $q$, updated throughout the entire training process can be greater than $(1 - s)\%$ of parameters updated. (i.e $q \geq 1 - s$). Analysing $q$ can reveal how much parameters are truly impactful for training, thereby addressing our second question.

We conducted this experiment using the same setup: finetuning the DistilBERT model on the GLUE-CoLA dataset after pre-training on the IMDb dataset. The results of this experiment are summarized

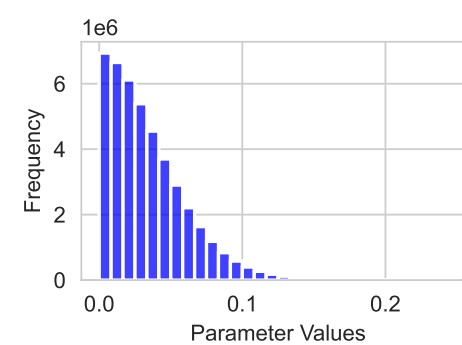
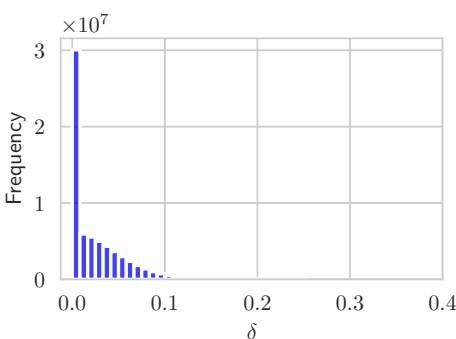

(a) Histogram of the parameters that changed during finetuning.

(b) Histogram of changes in parameter magnitudes, revealing that most changes are small.

Figure 3: Analysis of weight magnitudes of DistilBERT model pretrained on IMDb dataset and finetuned on CoLA dataset suggesting that finetuning predominantly affects a narrow set of impactful parameters. This supports our hypothesis that focusing updates on a smaller number of important parameters can yield efficient training.

in the Table 3. Additionally, we performed similar experiments on other GLUE (Wang et al., 2018) datasets, with the results presented in Section A.3

Our observations indicate that as $s$ decreases, that is when more parameters are updated per iteration, the number of unique parameters $q$ increases. This results in better model performance. *The increasing number of unique updates indicates that different parameters may become important at different stages of training, necessitating a more dynamic approach to parameter updates.*

We also evaluated the model with varying update frequencies $m$. Naturally, as $m$ increases, the model has fewer opportunities to update a larger number of parameters, leading to a decrease in $q$. Interestingly, there is an optimal point up to which performance either improves or remains stable, beyond which it begins to decline. This suggests that updating too many parameters or too few parameters can be detrimental to the training process. *These findings highlight the need for adaptive methods to determine the choice of parameters and the update frequency.*

The findings from our experiments suggest that a parameter-efficient training method that updates a small set of parameters each iteration is feasible. However, some key aspects require clarification:

**Parameter Selection Criteria**: Our analysis in Figure 3 indicates that the model frequently updates parameters with lower weight magnitudes. This contradicts the very premise of magnitude pruning. Therefore its not clear if adopting magnitude as a parameter importance criteria will help, in general. In our work, we bank on evidence from prior work on greedy parameter selection strategy Ramesh et al. (2023); Nutini et al. (2022) and use gradient as the parameter importance criteria.

**Parameter Selection Frequency**: As discussed earlier, determining when to change the set of parameters to update for a given training phase is crucial. In BlockLLM, we developed an adaptive strategy using the current loss value to decide when the parameter selection needs to be revisited.

## 2.2 BLOCKLLM

In this section we introduce BlockLLM, a parameter and memory efficient training method designed to reduce the number of trainable parameters in large language models (LLMs) without compromising training performance. Akin to other parameter-efficient fine-tuning (PEFT) methods such as LoRA (Hu et al., 2021) and ReLoRA (Lialin et al., 2023), BlockLLM updates only $k$ parameters at any iteration $t$. Here $k \ll n$ and $n$ is the total number of parameters. However, the main difference is that BlockLLM optimizes parameter selection by focusing on the most impactful parameters at different stages of the training process. The overall algorithm of BlockLLM is given in Algorithm 1.

**Parameter Selection Criteria.** In the context of LLMs, at iteration $t$, the update to the parameter is the processed gradient $\tilde{G}_t$, calculated using optimizers such as Adam (Kingma & Ba, 2014). Specifically, for any layer $l$ of the model, the update is given by,

$$\tilde{G}_t^l = M_t^l / \sqrt{V_t^l + \epsilon} \quad \text{where, } M_t^l = \beta_1 M_{t-1}^l + (1 - \beta_1)G_t^l, \quad V_t^l = \beta_2 V_{t-1}^l + (1 - \beta_2)G_t^{l\,2}. \quad (1)$$

All the operations in equation 1 are applied element-wise. Here, $\beta_1$ and $\beta_2$ are hyperparameters of the optimizer. $G_t^l \in \mathbb{R}^{p \times q}$ is the gradient at layer $l$. $M_t$ and $V_t$ denote the bias-corrected first moment estimate and bias-corrected second moment estimate respectively.

BlockLLM achieves memory savings by storing these optimizer states $M_t$ and $V_t$ only for the currently selected layers, rather than for all parameters in the model. When the set of selected layers changes, the optimizer is reset with these new layers. This means it no longer maintains the optimizer states for the previously selected layers, similar to methods such as ReLoRA Lialin et al. (2023). As an alternative, we also tried to offload $M_t$ and $V_t$ of the selected parameters to CPU and re-load them as needed. But that did not improve the model performance. Thus we decided to adopt the former strategy to avoid the offloading operation in the interest of faster training.

During the backward pass, BlockLLM selects the layers with large $||\tilde{G}_t^l||$ and updates only those layers. These selected layers are denoted as the set $S$. Note that by selecting full layers, we may not achieve the desired sparsity level $s$. Therefore, for each selected layer we construct a binary mask to retain only the top $k$ parameters by gradient magnitude:

$$\text{mask}[i, j] = \begin{cases} 1 & \text{if } |\tilde{G}_t^l[i, j]| \geq \tau \\ 0 & \text{otherwise,} \end{cases}$$

where $\tau$ is an estimated threshold, computed by looking at the gradient values of each layer. Specifically, $\tau$ is obtained computing the $(1 - \zeta)^{th}$ percentile in $\tilde{G}_t^l$. The value of $\zeta$ is defined as $\zeta = (\Sigma_p - n_s)/n_s$ (refer to Algorithm 2 for definitions of $\Sigma_p$ and $n_s$). Then, in every iteration $t$, the selected parameters in layers $l \in S$ are updated using the computed masks. The update rule is given by $W_{t+1}^l = W_t^l - \eta \left( \text{mask} \odot \tilde{G}_t^l \right)$, where $\eta$ is the learning rate. An illustration of the proposed parameter selection procedure is given in Figure 2.

However, there is one caveat with this approach. In the initial training iterations, the gradient estimates are known to be noisy. Additionally, in cases such as pretraining and finetuning with significant domain shifts, there is often very little useful inductive bias. Therefore, using gradients to select important parameters may prove to be detrimental to our cause in the initial few iterations.

To address this challenge, our experiments incorporate *layer visit frequency* $f$ into the selection criteria. Specifically, for any layer $l \in L = \{1, 2, \ldots, n\}$, then $f_l$ represents the sum normalized number of times the layer has been selected. That is,

$$S_t^l = \begin{cases} 1 & \text{if layer } l \text{ is selected at time } t \\ 0 & \text{otherwise.} \end{cases}$$

The *layer visit frequency* $f_l$ for layer $l$ after $T$ time steps is given by $f_l = \frac{1}{T} \sum_{t=1}^{T} S_t^l$. Consequently, the layer selection criterion is modified to $|\tilde{G}_l|/f_l$. This modification favors layers with high gradient norms while also giving priority to layers that have been selected less frequently in previous iterations. Our experimental results demonstrate that this refined criterion enhances performance.

**Parameter Selection Frequency.** The natural next question is how many iterations to update the parameters in the same set of layers $S$. BlockLLM addresses this by using the loss $\phi$ as a critical signal for determining when to change the parameter selection. Specifically, BlockLLM introduces a hyperparameter, patience $m$. At any iteration $t$, if $\phi_t$ equals to or exceeds the moving average of losses over the last $m$ iterations, the set $S$ is revised. The detailed parameter selection frequency algorithm is provided in Algorithm 2.

**Memory Efficiency** The memory benefits of training with BlockLLM, stems primarily from its parameter-efficient training approach. In practice, updating fewer parameters directly reduces the

number of gradients and optimizer states that need to be stored in VRAM. With $s\%$ sparsity, Block-LLM reduces the optimizer states by $s\%$ compared to full parameter training. Our empirical analysis corroborates this.

The parameter selection in BlockLLM relies on $\|\tilde{G}_l\|$ for each layer $l \in L$, which requires computing $\tilde{G}_l$ for all layers. This operation consumes significant memory required to store all the gradients. To reduce this, we sample a small number of $p$ additional layers per iteration besides the current block. Gradients for these $p$ layers are computed and their norms stored in a dictionary, which is then used for efficient parameter selection.

---

**Algorithm 1** BlockLLM Training Algorithm

1: **Input:** Data $X$, initial model parameters $W_0$, sparsity $s$, set of layers $L$, learning rate $\eta$, patience parameter $m$, $\beta_1$, $\beta_2$, $\epsilon$.
2: Initialize: $M_0 = 0, V_0 = 0, H = []$
   `//Forward and backward pass`
3: **for** each iteration $t$ **do**
4:     Loss $\phi_t$, $G_t$ = COMPUTE GRADIENT($W_t$)
5:     **if** (length($H$) $\geq m$ **and**
       $\phi_t \geq \frac{1}{m}\sum_{i=t-m+1}^{t} H[i]$) **or** $(t = 0)$ **then**
6:        mask, $S$ = SELECTPARAM($G_t$, $s$, $L$)
7:        $H = []$ `//Reset loss history`
8:     **end if**
9:     `// Update selected params`
10:     **for** each $l \in S$ **do**
11:        $G_t^l$ = COMPUTE GRADIENT($W_t^l$)
12:        $M_t^l, V_t^l, \tilde{G}_t^l$ = ADAM($M_{t-1}^l, V_{t-1}^l, G_{t-1}^l$)
13:        $\tilde{G}_t^l = \text{mask} \odot \tilde{G}_t^l$
14:        $W_{t+1}^l = W_t^l - \eta \tilde{G}_t^l$
15:     **end for**
16: **end for**

**Algorithm 2** Select Parameters Function

1: **function** SELECTPARAM($G_t$, $s$, $L$)
2:     Compute $\|\tilde{G}_t^l\|$ for each layer $l \in L$
3:     $n = \sum_{l \in L} \text{count}(l), n_s = (1-s) \times n$
4:     $D \leftarrow \text{sort}(L, \text{desc}, \|\tilde{G}_t^l\|/f_l)$
5:     Initialize $S = [], \Sigma_p = 0$
6:     **for** each $l \in D$ **do**
7:        $\Sigma_p \mathrel{+}= \text{count}(l)$ , $S \leftarrow S \cup \{l\}$
8:        **if** $\Sigma_p \geq n_s$ **then break**
9:     **end if**
10:     **end for**
11:     **for** each $l \in S$ **do**
12:        Let mask$_l = \mathbf{0}_{|G_t^l|}$
13:        **for** each $i, j$ in mask **do**
14:           **if** $\|\tilde{G}_t^l[i, j]\| \geq \tau$ **then**
15:              mask$_l[i, j] = 1$
16:           **end if**
17:        **end for**
18:     **end for**
19:     **return** mask, $S$
20: **end function**

Figure 4: (Left) The main BlockLLM algorithm, (right) SELECTPARAM function that performs parameter selection at iteration $t$.

## 3 EXPERIMENTS

We evaluated BlockLLM on both finetuning and pretraining tasks.[1] The large-scale finetuning experiments were conducted on an H100 GPU (80 GB), while the pretraining tasks were performed on NVIDIA A40 (48 GB) and A100 GPUs (80 GB) (one GPU allocated per experiment). The remaining experiments were conducted using a Tesla V100 GPU (32 GB).

### 3.1 LARGE SCALE FINETUNING

We conducted a series of experiments to evaluate the effectiveness of our approach in finetuning large language models. Specifically, we utilized the LLaMA-2 model (Touvron et al., 2023) with 7 billion parameters, and fine-tuned it on the Alpaca dataset (Peng et al., 2023). Alpaca dataset Peng et al. (2023) is a widely used benchmark for instruction following tasks. It consists of diverse instruction-response pairs across multiple domains. For the finetuning process, we employed the Llama-factory framework (Zheng et al., 2024). Llama-factory (Zheng et al., 2024) is an open-source framework designed for efficient fine-tuning, inference, and deployment of LLaMA models.

We finetuned the LLaMA-2 model (Touvron et al., 2023) with the experimental setup as detailed in the Appendix 9. Gradient checkpointing was enabled in all cases. We compared the performance of

---

[1] All our experiments were based on the code released by Zhao et al. (2024). We thank the authors for making their code publicly available and for clearly documenting their experimental setup.

BlockLLM against other state-of-the-art memory efficient training methods such as GaLore (Zhao et al., 2024), LoRA (Hu et al., 2021) and BAdam (Luo et al., 2024). We excluded Adam (Kingma & Ba, 2014) from the comparison as it exceeds the available memory on an 80GB GPU, with an estimated requirement of over 120GB. We evaluated all methods based on peak memory consumed during training in GB(Gigabytes), training time, and both training and evaluation loss. The results are provided in the Figure 5. The results indicate that BlockLLM converges to a lower loss value

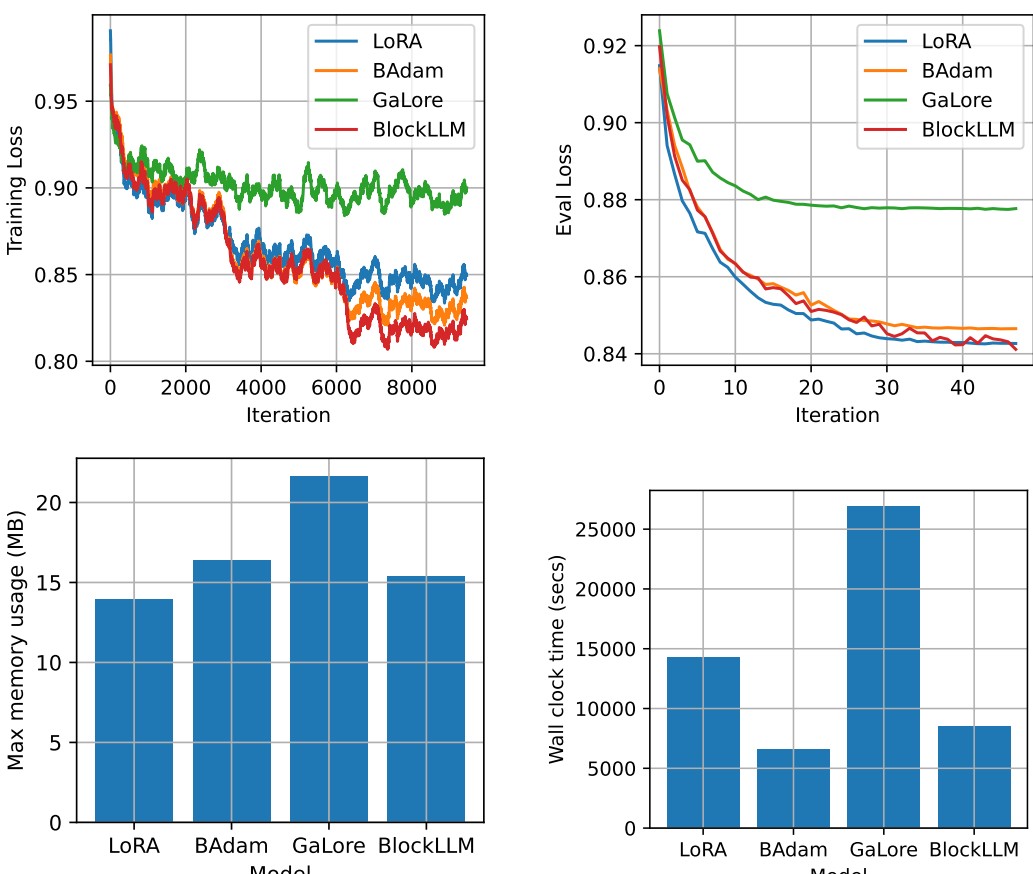

Figure 5: The figures, arranged from left to right and top to bottom, compare training loss, evaluation loss, peak memory usage, and training time for BlockLLM, LoRA, BAdam, and GaLore. BlockLLM demonstrates superior performance, with lower memory usage and reduced training time.

than the other methods, while requiring substantially less memory. Additionally, BlockLLM demonstrates strong generalization, achieving the lowest evaluation loss across all methods. In terms of runtime, BlockLLM performs comparably to BAdam and is faster than the other two baselines.

## 3.2 Pretraining on Llama model

We also compared BlockLLM with GaLore (Zhao et al., 2024) in pretraining LLaMA-based large language models Touvron et al. (2023) on the C4 (Colossal Clean Crawled Corpus) dataset (Raffel et al., 2020). The C4 dataset is a large-scale, cleaned version of the Common Crawl web corpus used for pre-training language models, featuring diverse and high-quality text from the internet. Our experiment setup is similar to Zhao et al. (2024), following the setup from Lialin et al. (2023). We finetuned the learning rate for BlockLLM while keeping all other hyperparameters fixed across experiments. We ran BlockLLM with the experimental setup described in A.7 and computed the perplexity scores from final evaluation loss and maximum memory usage in GB(Gigabytes). We ran GaLore for 10% of total iterations to observe memory consumption, and used the results from Zhao et al. (2024) for comparison. The perplexity and the memory are shown in Figure 6 and Table 1.

|  | 60M | | 130M | | 350M | |
|---|---|---|---|---|---|---|
|  | Perplexity | Memory | Perplexity | Memory | Perplexity | Memory |
| BlockLLM | **34.31** | **28.27** | **25.36** | **40.68** | 19.02 | **42.6** |
| GaLore | 34.88 | 32.26 | **25.36** | 46.69 | **18.95** | 49.06 |

Table 1: Comparison of BlockLLM's accuracy and VRAM memory usage (GB) for LLaMA models with GaLore. BlockLLM demonstrates reduced memory consumption while maintaining comparable performance.

We note here that BlockLLM takes a little bit longer to converge in some pretraining experiments ($130m$ and $350m$ models) compared to GaLore in the pretraining experiment. We suspect this is due to the fact that we are dealing with noisy gradients in the earlier iterations of training. However, BlockLLM converges to the state-of-the-art perplexity score in a few more iterations compared to GaLore. This issue is not present in the finetuning experiments.

**Effect of sparsity** $s$**.** Here, we compare BlockLLM with sparsity values $s = 0.5, 0.7$ and $0.9$ against GaLore (Zhao et al., 2024). The results are presented in Figure 6. We observe that with $s = 0.5$, BlockLLM consumes about $1.5$ GB less memory than Galore and higher sparsity values further reduce memory usage though this comes with the trade-off of requiring more training iterations for similar performance.

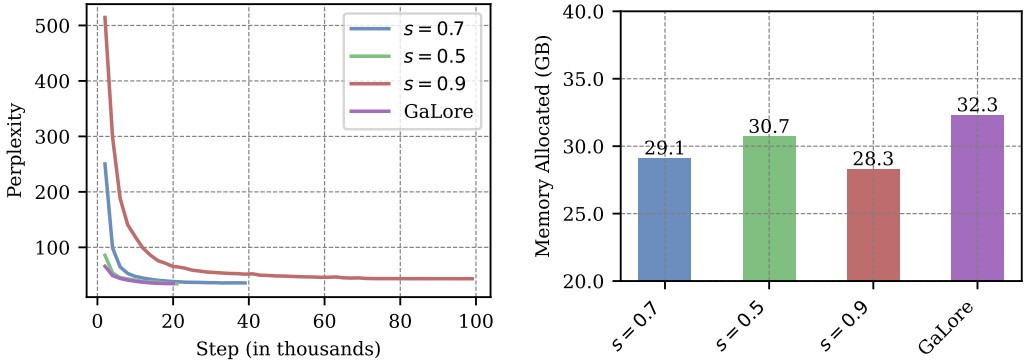

Figure 6: Comparison of perplexity (left) and memory usage (right) of Llama 60M. Here $s$ denotes the specified model sparsity. As it can be seen, BlockLLM performs competitively with GaLore, but at a much lower memory footprint.

### 3.3 ABLATION ON PARAMETER SELECTION STRATEGY

The parameter selection strategy is the cornerstone of our method, where we hypothesize that layers with large $||\tilde{G}_l||$ are important for training. To validate this, we conducted an ablation study in which we deliberately chose parameters with small $||\tilde{G}_l||$ (opposite to BlockLLM). Specifically, we finetuned LLaMA2 7B model on the Alpaca dataset, where parameters with the smallest gradient norms were selected for updates. We call this method **BLockLLM-SubOPT**. Naturally, we expect that this parameter selection strategy to be severely detrimental to the training process.

Let $L = \{l_1, l_2, \ldots, l_n\}$ represent the set of all layers. Now, for each layer $l \in L$, we computed the processed gradients $\tilde{G}_l$. We then sorted the layers in ascending order based on the their processed gradient norms $||\tilde{G}_l||$ and computed $||\tilde{G}_l||/f_l$. From this ordered list, we selected top $k$ layers with small gradient norms until the sparsity requirement $s$ is satisfied. Then, in each iteration $t$, the optimizer updates only the parameters in the selected $k$ layers. We conducted hyperparameter tuning and set $m = 100$ for $s = 0.95$. Now, we compared the training loss of both the methods and the comparative results are presented in Figure 7. As expected, the results show that BLockLLM-SubOPT exhibits significantly higher training loss and converges slower than BlockLLM.

**Effect of Layer Visit Frequency $f$.**  To evaluate this, we conducted a similar experiment on the LLaMA 60M model pretrained on the C4 dataset, where we assessed BlockLLM's strategy to select layers based on layer visit frequency $f$. To perform this, we selected layers solely based on their gradient norms without considering $f$. We compared this method with BlockLLM and the results are presented in Figure 7. Our hypothesis was that selecting parameters solely based on $||\tilde{G}||$ might result in higher loss during the initial iterations, followed by a gradual reduction in later iterations. This could be because of the noisy gradients early in training. Important layers might not be chosen resulting in higher loss compared BlockLLM. As expected, we observed similar behavior as shown in the figure 7.

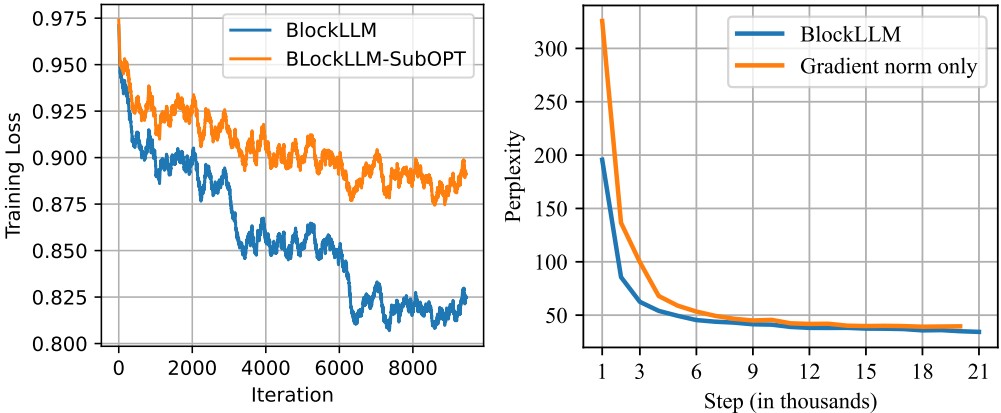

Figure 7: Ablation study on parameter selection criteria. The figure on the left illustrates the advantage of selecting parameters based on gradient norms and the figure on the right demonstrates the benefit of incorporating layer selection frequency.

## 4  CONCLUSION & FUTURE WORK

In this paper we introduced BlockLLM, a novel method for efficiently training large language models. By dynamically estimating and updating the importance of parameters during training, BlockLLM effectively achieves state-of-the-art performance while significantly reducing the memory footprint. Our method achieves the highest validation accuracy on the finetuning tasks, sometimes even surpassing full finetuning. One key aspect of BlockLLM is that it does not presuppose the importance of layers but continuously evaluates and updates parameter importance throughout training. This adaptive approach allows for more flexible and efficient optimization compared to methods that assume certain parameters are critical from the outset. Additionally, BlockLLM preserves the original architecture without altering the model structure or restricting the parameter search space, making it suitable for various LLMs and tasks.

**Broader Impacts** Our work aims to reduce the memory and computational requirements of training LLMs. First, our technology democratizes access to LLM training, making it more feasible for student researchers and institutions with limited computational resources to participate in cutting-edge AI research. Furthermore, the low-memory requirements of our method means that one can train with larger batch sizes and achieve faster convergence. This has a direct effect on the environment.

**Future works.** Future work on BlockLLM could explore several promising avenues. Currently, our research has focused on parameter selection based on gradient norms, but BlockLLM can be seen as a framework for parameter-efficient training rather than a single algorithm. This opens the door to investigating alternative criteria for parameter selection, potentially tailored to specific problems or tasks. Moreover, while our ablation studies on BlockLLM's hyperparameters have provided insights into their impact on training, further research is needed to understand how different layers might be affected by greedy parameter selection strategies. BlockLLM also complements existing memory-optimized training techniques, including those discussed in this paper. Exploring the integration of BlockLLM with methods like quantization or GaLore could further reduce memory consumption.

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

# A APPENDIX

## A.1 SPARSITY ACCURACY TRADEOFF

We performed magnitude pruning on the IMDb pre-trained model (Maas et al., 2011) weights at various sparsity levels and fine-tuned these pruned models on the GLUE-CoLA dataset (Wang et al., 2018). The results of these experiments, detailing the relationship between sparsity and accuracy, are summarized in Table 2.

| Sparsity | Accuracy |
|:--------:|:--------:|
| 0.0 | 79.57 |
| 0.5 | 78.52 |
| 0.6 | 74.2 |
| 0.7 | 67.68 |
| 0.8 | 69.12 |
| 0.9 | 69.12 |

Table 2: Performance of pruned models at various sparsity levels on GLUE-CoLA dataset.

The accuracy generally declines with increasing sparsity. At 0.5 sparsity, the performance remains relatively high at 78.52%, close to the non-pruned model's 79.57%. However, accuracy drops more significantly to 67.68% at 0.7 sparsity. Interestingly, at sparsity levels of 0.8 and 0.9, accuracy stabilizes around 69.12%.

## A.2 ANALYSIS OF WEIGHT MAGNITUDES

In this experiment, we pretrain DistilBERT on the IMDB dataset (Maas et al., 2011) and then fine-tune it on GLUE-CoLA (Wang et al., 2018) with sparsity $s = 0.7$. We then plot the histogram of the weight magnitudes $W^t$ where $\delta = |w_i^0 - w_i^t| > \eta$, with $\eta$ as the threshold. We set $\eta = 0.001$ in this case.

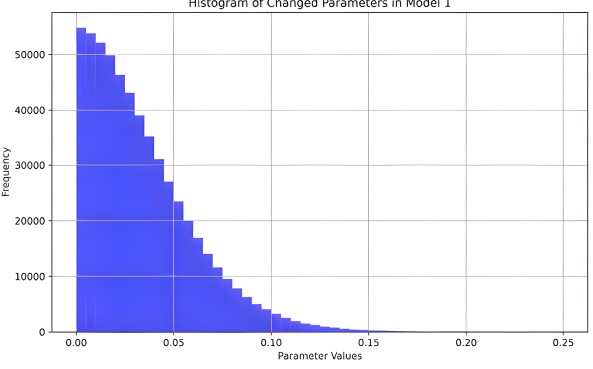

Figure 8: Histogram of changed parameters for $s = 0.7$

## A.3 ANALYSIS OF REDUCED PARAMETER TRAINING

In this experiment, we updated the chosen parameter set $S$ every $m$ iterations based on the weight magnitudes $|W^t|$, at current iteration $t$. This means that after every $m$, the parameter selection criteria is revisited to obtain a new set of parameters to update. The objective is to understand if $S$ changes significantly over time and if adaptively selecting $S$ enhances the training performance. In this framework, we continue to update only the top $k = n \times (1 - s)$ parameters in each iteration. However, the percentage of unique parameters $q$, updated throughout the entire training process can

be greater than $(1-s)\%$ of parameters updated. (i.e $q \geq 1-s$). Analysing $q$ can reveal how much parameters are truly impactful for training, thereby addressing our second question.

We conducted this experiment using the same setup: finetuning the DistilBERT model on the GLUE-CoLA dataset after pretraining on the IMDb dataset. The results of this experiment are summarized in the Table 3.

| $1-s$ | $q$ | $m$ | **Accuracy** | **Matthews Correlation** |
|---|---|---|---|---|
| 0.1 | 0.58 | 1000 | 82.55 | 0.5882 |
| 0.02 | 0.36 | 1000 | 82.45 | 0.5711 |
| 0.02 | 0.14 | 4000 | 82.45 | 0.5794 |
| 0.02 | 0.10 | 6000 | 82.07 | 0.5679 |

Table 3: Impact of Update Frequency and Sparsity on CoLA Dataset

Next, we fine-tuned the GLUE datasets (Wang et al., 2018) on the DistilBERT model (Sanh et al., 2019) pretrained on the IMDb dataset (Maas et al., 2011). We varied the sparsity $s$ and update frequency $m$ while monitoring the number of unique parameters updated $q$. We ran the GLUE-SST2 experiments for 8400 iterations and GLUE-STSB for 20000 iterations. Additionally, we tracked the VRAM usage for the GLUE-SST2 dataset to compare it with the memory consumption of full-parameter fine-tuning, which is 7.9 GB. This comparison aims to determine if reduced parameter training effectively decreases memory usage.

| $1-s$ | $q$ | $m$ | Spearman Correlation |
|---|---|---|---|
| 0.01 | 0.05 | 5000 | 88.82 |
| 0.01 | 0.037 | 10000 | 88.77 |

Table 4: Impact of Update Frequency and Sparsity on STSB Dataset

| $1-s$ | $q$ | $m$ | Accuracy | VRAM |
|---|---|---|---|---|
| 0.008 | 0.04 | 2400 | 91.97 | 4.4 |
| 0.01 | 0.11 | 3000 | 90.94 | 5.5 |
| 0.02 | 0.13 | 1000 | 90.59 | 5.5 |
| 0.02 | 0.16 | 2000 | 92.2 | 5.5 |

Table 5: VRAM Usage with Different Update Frequencies on SST2 Dataset

Table 4 shows the impact of update frequency $m$ and sparsity $s$ on the STSB dataset, where the correlation remains stable despite changes in update frequency. As $m$ increased too high, the performance declined.

## A.4 VRAM MEMORY

All memory values presented in our tables represent actual observed memory usage in gigabytes (GB) rather than estimates. Memory consumption was monitored using the "nvidia-smi" command, and the maximum memory usage recorded during the training process was noted.

## A.5 FINETUNING ON GLUE

The General Language Understanding Evaluation (GLUE) benchmark (Wang et al., 2018) is widely used to evaluate the performance of NLP models across a range of tasks, including sentiment analysis, question answering, and textual entailment. We benchmarked the performance of BlockLLM against GaLore (Zhao et al., 2024) and full finetuning (FFT) using the pre-trained RoBERTa model

(Liu et al., 2019) on GLUE tasks. We did not compare against LoRA Hu et al. (2021) and its variants Lialin et al. (2023); Kamalakara et al. (2022), because our experimental setup aligns with those described in ReLoRA and GaLore(Lialin et al., 2023; Zhao et al., 2024). The results for these methods are already documented in GaLore(Zhao et al., 2024).

For BlockLLM, we conducted hyperparameter tuning for the learning rate and the learning rates for the different tasks are as follows. The batch size was set to 32 for the CoLA dataset, and 16 for all

Table 6: Hyperparameter details for the GLUE experiments

|               | MRPC   | COLA   | STS-B  | RTE    | SST2   | MNLI   | QNLI   | QQP    |
|---------------|--------|--------|--------|--------|--------|--------|--------|--------|
| Learning rate | 3E-05  | 5E-05  | 3E-05  | 3E-05  | 3E-05  | 3E-05  | 1E-05  | 3E-05  |

other datasets. For all tasks, we used $s = 0.95$ and $m = \frac{1}{4} \times$ total number of iterations. VRAM memory usage was monitored and recorded as described in Section A.4. For GaLore, we used the learning rate specified in their original work (Zhao et al., 2024). In our experiments, we used $s = 0.95$. We evaluated both performance and memory consumption during the training process for all methods. The results, presented in Tables 7 and 8, indicate that BlockLLM outperforms the other models in all tasks while achieving approximately a 13.5% reduction in memory usage on average.

|                 | MRPC  | COLA | STS-B | RTE   | SST2 | MNLI  | QNLI  | QQP  | Avg. |
|-----------------|-------|------|-------|-------|------|-------|-------|------|------|
| Block-LLM       | **3.97** | **2.8** | **3.48** | **9.1** | **3.6** | **13.7** | **12.8** | 8.36 | **7.2** |
| GaLore (rank=8) | 4.52  | 4.2  | 4.8   | 9.7   | 3.87 | 15.1  | 14.8  | 8.43 | 8.18 |
| GaLore (rank=4) | 4.52  | 4.2  | 4.8   | 9.7   | 3.86 | 15.2  | 14.8  | **8.03** | 8.14 |
| FFT             | 4.24  | 3.67 | 4.4   | 10.28 | 3.82 | 15.53 | 15.02 | 9.22 | 8.27 |

Table 7: VRAM Memory Comparison Across Different Tasks (measured in GB). VRAM memory usage was monitored as described in Section A.4.

|                 | MRPC  | COLA   | STS-B | RTE   | SST2  | MNLI  | QNLI  | QQP   | Avg.  |
|-----------------|-------|--------|-------|-------|-------|-------|-------|-------|-------|
| Block-LLM       | 91.8  | **63.8** | 90.02 | 80.14 | **94.95** | **87.75** | 92.95 | 91.36 | **86.6** |
| GaLore (rank=8) | 89.96 | 62.5   | **91.1** | 79.78 | 94.38 | 87.17 | **92.97** | 91.11 | 86.12 |
| GaLore (rank=4) | 91.7  | 61.67  | 91.09 | 79.78 | 94.04 | 87    | 92.65 | 91.06 | 86.12 |
| FFT             | **92.36** | 62.84 | 91.1 | **80.5** | 94.57 | 87.18 | 92.33 | **92.28** | **86.6** |

Table 8: Score Comparison Across Different GLUE Tasks

## A.6 LARGE SCALE FINETUNING OF LLAMA 2 ON ALPACA

We provide more details on the hyperparameters used in training the Llama 2 model Touvron et al. (2023) on Alpaca datasetPeng et al. (2023).

| Hyperparameter | BlockLLM | GaLore | LoRA | BAdam |
|---|---|---|---|---|
| Learning Rate (LR) | $1 \times 10^{-5}$ | $1 \times 10^{-5}$ | $1 \times 10^{-5}$ | $1 \times 10^{-5}$ |
| LR Scheduler | Cosine (lr_min = 0) | Cosine (lr_min = 0) | Cosine (lr_min = 0) | Cosine (lr_min = 0) |
| Epochs | 3 | 3 | 3 | 3 |
| Batch Size | 8 | 8 | 8 | 8 |
| Gradient Accumulation | 2 | 2 | 2 | 2 |
| Weight Decay | 0.01 | 0.01 | 0.01 | 0.01 |
| Sparsity ($s$) | 0.95 | NA | NA | NA |
| Patience(m) | 100 | NA | NA | NA |
| K | NA | NA | NA | 100 |
| rank | NA | 8 | 8 | NA |
| $\alpha$ | NA | 2 | $4 \times rank$ | NA |

Table 9: Hyperparameter Details for finetuning LLaMA 2 Alpaca dataset

## A.7 PRETRAINING ON LLAMA

We present the hyperparameters utilized for training the LLama models with sizes 60M, 130M and 350M in 10. For 60M and 130M experiments, the maximum sequence length was set to 256 with a gradient accumulation of 2,and for 350M with a batch size of 128 with a gradient accumulation of 4. A cosine annealing schedule was employed for learning rate adjustment, decaying to 10% of the initial learning rate. For BlockLLM, no learning rate warmup was applied. However, for GaLore, the learning rate was warmed up for the first $10\%$ of training, following the approach outlined in Zhao et al. (2024). The parameter $m$ was set to 50 for all the experiments.

| | 60M | 130M | 350M |
|---|---|---|---|
| Learning rate | 1E-03 | 1E-03 | 1E-03 |
| Total training steps | 10K | 20K | 60K |
| $s$ | 0.5 | 0.5 | 0.5 |

Table 10: Hyperparameter Details for Pretraining LLaMA Models with BlockLLM on the C4 Dataset

## A.8 ABLATION ON THE HYPERPARAMETER $m$

We investigated the sensitivity of the model to the patience parameter $m$ in both fine-tuning and pre-training setups. These experiments were conducted using the GLUE benchmark and the LLaMA 2 model on the C4 dataset. Throughout the experiments, we fixed all parameters of the Adam optimizer and maintained a sparsity level of $s = 0.5$ while varying $m$. The results are presented in Figure 9. Our observations indicate that in the fine-tuning setting, the model is relatively insensitive to variations in $m$. Specifically, setting $m = 50$ or $m = 1000$ did not result in significant performance differences. This finding aligns with the observations reported in Zhao et al. (2024), which suggest that gradients change more slowly. The gradual nature of gradient changes implies a correspondingly gradual variation in the optimal parameter set, thereby reducing sensitivity to changes in $m$. In contrast, in the pre-training setting, smaller values of $m$ lead to faster convergence. This behavior can be attributed to the presence of noisy gradients in the earlier iterations of pre-training. Consequently, a smaller $m$ helps maintain impactful parameter selection particularly in the initial phase of training, thereby facilitating faster convergence.

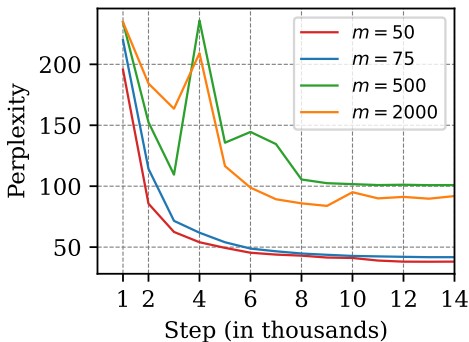

Figure 9: Ablation on the patience parameter $m$. We see that $m$ affects the algorithm and model performance significantly in our pre-training experiments.

