# OpenReview forum: "BlockLLM: Memory-Efficient Adaptation of LLMs by Selecting and Optimizing the Right Coordinate Blocks"
_ICLR.cc/2025/Conference — ICLR 2025 Conference Withdrawn Submission_

### Official Review · Reviewer_kKPk · 2024-11-02

**Soundness:** 2
**Presentation:** 2
**Contribution:** 2
**Rating:** 3
**Confidence:** 4

**Summary:**

This paper proposes a memory-efficient training approach based on Block coordinate descent. They selectively choose a few layers and parameters to update during training, achieving memory efficiency and good performance.

**Strengths:**

(1) This paper focuses on an important research question, memory-efficient training.

(2) BlockLLM achieves better performance than GaLore in pre-training.

**Weaknesses:**

(1) The references in this paper are not cited properly. Citep and citet are used randomly.

(2) Pruning is presented as one of the existing strategies for memory-efficient training in the introduction. IMHO, pruning is a post-training method, even though sometimes re-training is required.

(3) Can the authors explain the meaning of altering the training dynamics in line 79?  I assume that BlockLLM also has a different training dynamic with full-parameter training.

(4) Can the authors explain what "This suggests that there is some inductive bias that the model can leverage when finetuning on a dataset with significant domain shift, under a reduced parameter setting" means exactly? I found this claim ambiguous.

(5) What does unique parameter $q$ mean in the paper?

(6) One major concern is that the main analysis in this paper is fine-tuning Bert models on GLUE tasks, which is very different from the Llama pre-training scenario that the baselines in this paper focus on such as GaLore and BAdam. I don't think the observation in section 2 has a strong correlation to the later methodology and the result part. More importantly, the intuition obtained from pruning and fine-tuning is also a different scenario than pre-training LLMs.

(7) It is not very obvious to me how the layers are selected during training.

(8) The maximum model size is 350M, which is not a large language mode nowadays, to be honest.

(9) The comparison with Badam is missing in Table 1, which is also a BCD approach.

**Questions:**

(1) The references in this paper are not cited properly. Citep and citet are used randomly.

(2) Pruning is presented as one of the existing strategies for memory-efficient training in the introduction. IMHO, pruning is a post-training method, even though sometimes re-training is required.

(3) Can the authors explain the meaning of altering the training dynamics in line 79?  I assume that BlockLLM also has a different training dynamic with full-parameter training.

(4) Can the authors explain what "This suggests that there is some inductive bias that the model can leverage when finetuning on a dataset with significant domain shift, under a reduced parameter setting" means exactly? I found this claim ambiguous.

(5) What does unique parameter $q$ mean in the paper?

(6) One major concern is that the main analysis in this paper is fine-tuning Bert models on GLUE tasks, which is very different from the Llama pre-training scenario that the baselines in this paper focus on such as GaLore and BAdam. I don't think the observation in section 2 has a strong correlation to the later methodology and the result part. More importantly, the intuition obtained from pruning and fine-tuning is also a different scenario than pre-training LLMs.

(7) It is not very obvious to me how the layers are selected during training.

(8) The maximum model size is 350M, which is not a large language mode nowadays, to be honest.

(9) The comparison with Badam is missing in Table 1, which is also a BCD approach.

---

### Official Review · Reviewer_TwSW · 2024-11-02

**Soundness:** 3
**Presentation:** 3
**Contribution:** 3
**Rating:** 5
**Confidence:** 3

**Summary:**

This paper presents BlockLLM, a memory-efficient method for fine-tuning and pretraining LLMs by selectively updating a subset of parameters. Inspired by block coordinate descent, BlockLLM achieves state-of-the-art performance with significantly reduced memory consumption, particularly advantageous in resource-limited settings.

**Strengths:**

1. The paper proposes a novel approach to reduce memory footprint in training LLMs, enabling broader access to model training, especially under limited resources.
2. Experiments validate BlockLLM's effectiveness, showing lower memory usage and comparable or superior performance across multiple benchmarks. The ablation studies and comparison with other state-of-the-art methods (like LoRA and GaLore) provide a robust evaluation of the approach’s efficacy.

**Weaknesses:**

1. In terms of fine-tuning results, while there are some advantages in training loss compared to existing methods, the evaluation loss remains similar to—or even worse than—that of LoRA. Typically, evaluation results are more representative than training loss in fine-tuning.
2. Regarding pretraining results, the improvements over GaLore are still incremental. Additionally, the model and data scale are too small to show meaningful information. And it would also be helpful to place full parameter pretraining baseline as a reference.

**Questions:**

I may have overlooked this in the paper, but when updating the selected parameters, how are the Adam states—specifically the momentum and variance terms—handled?

---

### Official Review · Reviewer_3twK · 2024-11-03

**Soundness:** 1
**Presentation:** 3
**Contribution:** 2
**Rating:** 3
**Confidence:** 4

**Summary:**

This paper introduces BlockLLM, a memory-efficient approach for training large language models. In this method, only a small subset of parameters with the highest top-k gradient norms are unfrozen, reducing the number of trainable parameters and minimizing memory usage. Furthermore, BlockLLM dynamically adjusts the selected parameter set via periodic monitoring of the loss, along with an additional hyperparameter of layerwise visit frequency $f$ to counteract the estimation error incurred by gradient noise.

Experimental results on LLaMA-2 + Alpaca on instruction-following tasks and a small llama-based LLM + C4 on pretraining tasks demonstrate that BlockLLM achieves lower or on-par loss and perplexity, with memory consumption lower or at least comparable to GaLore and other baselines.

**Strengths:**

This paper is well-written and easy to follow. In particular, the motivation from magnitude pruning and discussion of Galore's limitations are intriguing, where further analysis in depth would be interesting. The proposed BlockLLM algorithm also provides one more memory-efficient training option for the LLM community, allowing model pretraining to be more accessible for researchers and engineers with limited resources.

**Weaknesses:**

The main weakness of this paper is its quality. BlockLLM's state-of-the-art (SOTA) performance is certainly an overclaim given the presented results in the experimental section, where the chosen benchmarks, model & dataset, and baselines are all far from enough to support the SOTA claim.

To compensate for this deficiency, I would suggest,
  * 1) **Benchmarks**: Utilize standard/conventional benchmarks for the fine-tuning & pretraining tasks, instead of just presenting the loss and perplexity.
    - For fine-tuning tasks, there are several options, where at least three of them should be included to claim SOTA performance
      + Instruction-following (e.g. for Alpaca): MT-Bench [1], IFEval [2]
      + Math: GSM8K [3], Math [4]
      + Coding: HumanEval [5], MBPP [6]
      + Other: AGIEval [7], Winogrande [8]

    - For pre-training tasks, the following tasks can also be utilized, where at least three of them should be included to claim SOTA performance
      + Knowledge: MMLU [9], MMLU-Pro [10]
      + Commonsense reasoning: ARC-Easy & ARC-Challenge [11], BoolQ [12], OpenBookQA [13], PIQA [14], SIQA [15], HellaSwag [16].

  * 2) **Models & Datasets**:
    - For model, utilize SOTA open-source models such as llama-3 [17] instead of the old llama-2, where later versions like llama-3.1, llama-3.2 are more preferable. Other models such as Qwen2.5 [18], and Gemma-2 [19] can further demonstrate the effectiveness of the proposed method. At least two different families of models are recommended to be included to show the methods' applicability across different model choices.
    - For datasets, utilize better datasets such as OpenHermes [20].

  * 3) **Baselines**: at least two more memory-efficient methods to be compared with, such as HFT [21], LISA [22], and WeLore [23].

### Other detailed comments:
  - Algorithm 2: better specify $\tau$ in the input/arguments, or add the corresponding computation procedure for $\tau$ from $s$
  - line 376: typo: Appendix 9 -> Table 9

### Reference
[1]: Zheng, Lianmin, et al. "Judging llm-as-a-judge with mt-bench and chatbot arena." Advances in Neural Information Processing Systems 36 (2023): 46595-46623.

[2]: Zhou, Jeffrey, et al. "Instruction-following evaluation for large language models." arXiv preprint arXiv:2311.07911 (2023).

[3]: Cobbe, Karl, et al. "Training verifiers to solve math word problems." arXiv preprint arXiv:2110.14168 (2021).

[4]: D. Hendrycks, C. Burns, S. Kadavath, A. Arora, S. Basart, E. Tang, D. Song, and J. Steinhardt. Measuring Mathematical Problem Solving With the MATH Dataset. In Neural Information Processing Systems: Datasets and Benchmarks, 2021.

[5]: Chen, Mark, et al. "Evaluating large language models trained on code." arXiv preprint arXiv:2107.03374 (2021).

[6]: Austin, Jacob, et al. "Program synthesis with large language models." arXiv preprint arXiv:2108.07732 (2021).

[7]: Zhong, Wanjun, et al. "Agieval: A human-centric benchmark for evaluating foundation models." arXiv preprint arXiv:2304.06364 (2023).

[8]: Sakaguchi, Keisuke, et al. "Winogrande: An adversarial winograd schema challenge at scale." Communications of the ACM 64.9 (2021): 99-106.

[9]: Hendrycks, Dan, et al. "Measuring massive multitask language understanding." arXiv preprint arXiv:2009.03300 (2020).

[10]: Wang, Yubo, et al. "Mmlu-pro: A more robust and challenging multi-task language understanding benchmark." arXiv preprint arXiv:2406.01574 (2024).

[11]: Peter Clark, Isaac Cowhey, Oren Etzioni, Tushar Khot, Ashish Sabharwal, Carissa Schoenick, and Oyvind Tafjord. 2018. Think you have solved question answering? try arc, the ai2 reasoning challenge. ArXiv, abs/1803.05457.

[12]: Clark, Christopher, et al. "BoolQ: Exploring the surprising difficulty of natural yes/no questions." arXiv preprint arXiv:1905.10044 (2019).

[13]: Mihaylov, Todor, et al. "Can a suit of armor conduct electricity? a new dataset for open book question answering." arXiv preprint arXiv:1809.02789 (2018).

[14]: Bisk, Yonatan, et al. "Piqa: Reasoning about physical commonsense in natural language." Proceedings of the AAAI conference on artificial intelligence. Vol. 34. No. 05. 2020.

[15]: Sap, Maarten, et al. "Socialiqa: Commonsense reasoning about social interactions." arXiv preprint arXiv:1904.09728 (2019).

[16]: Zellers, Rowan, et al. "Hellaswag: Can a machine really finish your sentence?." arXiv preprint arXiv:1905.07830 (2019).

[17]: Dubey, Abhimanyu, et al. "The llama 3 herd of models." arXiv preprint arXiv:2407.21783 (2024).

[18]: Qwen Team. https://qwenlm.github.io/blog/qwen2.5/

[19]: Team, Gemma, et al. "Gemma 2: Improving open language models at a practical size." arXiv preprint arXiv:2408.00118 (2024).

[20]: Teknium. Openhermes dataset, 2023a. URL https://huggingface.co/datasets/
teknium/openhermes.

[21]: Hui, Tingfeng, et al. "HFT: Half Fine-Tuning for Large Language Models." arXiv preprint arXiv:2404.18466 (2024).

[22]: Pan, Rui, et al. "LISA: Layerwise Importance Sampling for Memory-Efficient Large Language Model Fine-Tuning." arXiv preprint arXiv:2403.17919 (2024).

[23]: Jaiswal, Ajay, et al. "From galore to welore: How low-rank weights non-uniformly emerge from low-rank gradients." arXiv preprint arXiv:2407.11239 (2024).

**Questions:**

I am wondering if there are recommended choices of $f$ and $m$ in Algorithm 1 for different models, as hyperparameter-tuning can be the main engineering effort for employing this BlockLLM in practice.

---

### Official Review · Reviewer_17UK · 2024-11-03

**Soundness:** 3
**Presentation:** 2
**Contribution:** 2
**Rating:** 5
**Confidence:** 3

**Summary:**

This paper introduces BlockLLM, a novel method inspired by block coordinate descent to address the significant memory challenges in training large language models (LLMs). Unlike existing approaches like LoRA and GaLore, BlockLLM carefully selects and updates a very small subset of trainable parameters without altering the model's architecture or training procedure. This method achieves state-of-the-art performance in both finetuning and pretraining tasks while significantly reducing the memory footprint of the optimization process.

**Strengths:**

- Interesting topic
- The paper is well-organized
- Storyline is completed

**Weaknesses:**

Please see the questions

**Questions:**

- The phenomenon of a sudden drop in performance under specific sparsity isn't new and has been explored in previous studies [1]

- The finding that significant updates are concentrated in a small subset of weights has been previously studied in [2]

- It's not entirely clear how the analysis of magnitude pruning leads to the development of the BlockLLM methodology. A more detailed explanation would help readers understand the logical progression from the analysis to the proposed method.

-  In Figure 4, the evaluation loss for BlockLLM appears higher than that of LoRA for most of the training period. This raises questions about the claim that BlockLLM converges to a lower loss value than other methods. Providing additional evidence or clarification could make this claim more convincing.

- The pretraining experiments are conducted on models with up to 350 M parameters. Conducting experiments on larger models, at least at the 1B level, would enhance the credibility of the results

[1] The Emergence of Essential Sparsity in Large Pre-trained Models: The Weights that Matter [1]

[2] Pruning Small Pre-Trained Weights Irreversibly and Monotonically Impairs “Difficult" Downstream Tasks in LLMs [2]

---

### Official Review · Reviewer_Vbpd · 2024-11-08

**Soundness:** 3
**Presentation:** 2
**Contribution:** 3
**Rating:** 3
**Confidence:** 3

**Summary:**

In this work, the authors introduce BlockLLM, a novel approach for memory-efficient LLM pre-training and adaptation. The method achieves efficiency by selectively updating only a small subset of trainable parameters. BlockLLM demonstrates superior performance compared to Galore in both computational efficiency and memory usage. This improvement stems from two key factors: first, BlockLLM avoids the computational overhead of SVD calculations that slow down Galore; second, BlockLLM's architecture allows it to be applied across all LLM layers, resulting in significantly reduced memory requirements. Experimental results on both fine-tuning and pre-training tasks show that BlockLLM achieves state-of-the-art performance while maintaining these efficiency advantages.

**Strengths:**

1. Novelty: This work introduces a block-sparsity approach for memory-efficient pre-training, leveraging an innovative selection strategy to determine optimal sparse patterns.

2. Computational Efficiency: The proposed method demonstrates significant improvements in both training speed and memory consumption compared to Galore.

3. Comprehensive Evaluation: Extensive empirical studies across both pre-training and fine-tuning stages, spanning large and small-scale models, validate the method's effectiveness and generalizability.

**Weaknesses:**

1. Title: The emphasis on pre-training and comparisons with Galore suggests that "adaptation" may not be the most accurate term in the title. Consider revising the title to better reflect the paper's core focus and contributions.

2. Fine-tuning Results: While BlockLLM demonstrates improved wall-clock time compared to LoRA with comparable evaluation loss, there are important trade-offs to consider. The maximum memory consumption of BlockLLM exceeds that of LoRA. Additionally, LoRA's architecture enables scalable serving scenarios (e.g., S-LORA [1]). BlockLLM's approach of modifying all parameters after many selection iterations, will reduce the maximum of loaded adapters with same memory. Additionally, even with one single iteration. the unstructured nature will slow down inference. Therefore, the authors should either emphasize more on the pre-training part or provide more evidence for the fine-tuning advantages.

3. Missing Explanation: From the introduction and abstract, the authors mention that BlockLLM achieves better memory efficient than Galore as it can be applied to layers without "reversibility" property. However, this work lack an explanation for this property. Additionally, it would be better if the authors can provide a detailed break down on the LLaMA structure to show which part of memory is saved. From my current understanding, I think it should save activation memory for the non-selected parameters?

4. Select Parameters Function: Here, the authors mention that we use the gradient to select layers and parameters. However, it is unclear that whether the selection process requires full gradients and lead to larger memory. If so, the authors should explain how they optimize this for better memory.

5. For Table 1, with the increased model size, Galore performs better than BlockLLM. I was wondering whether the authors can explain this part. Additionally, Galore shows faster coverage in Figure 6. Therefore, I think the authors should add a comparison between Galore and BlockLLM with same max memory requirements to show which one coverage faster and better.

[1] Sheng, Ying, et al. "SLoRA: Scalable Serving of Thousands of LoRA Adapters." Proceedings of Machine Learning and Systems 6 (2024): 296-311.

**Questions:**

See weaknesses.

---

### Note · Authors · 2024-11-22

**Comment:**

Thank you reviewers for taking time to review our paper. We have decided to withdraw our submission.

**Withdrawal Confirmation:**

I have read and agree with the venue's withdrawal policy on behalf of myself and my co-authors.